# NESTED POLICY REINFORCEMENT LEARNING

## ABSTRACT

Off-policy reinforcement learning (RL) has proven to be a powerful framework for guiding agents' actions in environments with stochastic rewards and unknown or noisy state dynamics. In many real-world settings, these agents must operate in multiple environments, each with slightly different dynamics. For example, we may be interested in developing policies to guide medical treatment for patients with and without a given disease, or policies to navigate curriculum design for students with and without a learning disability. Here, we introduce nested policy fitted Q-iteration (NFQI), an RL framework that finds optimal policies in environments that exhibit such a structure. Our approach develops a nested $Q$-value function that takes advantage of the shared structure between two groups of observations from two separate environments while allowing their policies to be distinct from one another. We find that NFQI yields policies that rely on relevant features and perform at least as well as a policy that does not consider group structure. We demonstrate NFQI's performance using an OpenAI Gym environment and a clinical decision making RL task. Our results suggest that NFQI can develop policies that are better suited to many real-world clinical environments.

## 1 INTRODUCTION

Off-policy reinforcement learning (RL) has proven to be a powerful framework for guiding agents' actions in environments with stochastic rewards and unknown or noisy state dynamics (13; 2). Previous off-policy RL algorithms (9) learn a *single* optimal policy designed to maximize agents' long-term expected rewards. However, such a learning scheme is not always ideal. In many real-world settings, agents' environments can be divided into *multiple* groups for the same task. These groups may share rewards but have different state dynamics, which often entails distinct optimal policies.

For example, say that we are designing a clinical decision support algorithm to treat ischemic stroke in two groups of patients: one group has a large number of patients with healthy kidneys and the other has a smaller number of patients with renal (kidney) disease. Note that there are similarities in the treatment plans for these two groups; in particular, both sets of patients exhibit the effects of disturbed blood flow to the brain due to stroke, and may require similar procedures. However, patients with renal disease must follow a treatment plan that considers the inability of their kidneys to properly filter out toxins. Traditional off-policy RL algorithms would learn a single policy that does not consider the underlying presence of renal disease. This policy is likely to overfit to the patients with functioning kidneys and overlook any known differences in optimal treatment plans. In such scenarios, it would be beneficial to learn distinct policies for pre-defined groups of patients, while leveraging any shared environmental structure and reward functions between the groups to ensure a large and effective sample size.

To address this problem, we introduce nested policy fitted Q-iteration (NFQI), an off-policy RL algorithm that extends fitted Q-iteration (FQI) (9) to account for nested environments. Here, *nested* refers to an environment structured with two predefined groups that share some characteristics; we will refer to the two groups as *background* and *foreground*. The background group often contains far more samples. In the clinical example above, the background group includes patients with healthy kidneys and the foreground group includes renal disease patients. Both groups share a reward function, in this case successfully treating a stroke, but have different state dynamics.

The contributions of this paper to off-policy RL in the presence of nested environments are threefold:

1. We introduce NFQI, a supervised learning method that uses a nested $Q$-value function to learn policies that are suited to nested environments. NFQI concurrently models both the background and foreground datasets and is agnostic to modeling choices.

2. We develop a training procedure inspired by transfer learning (31) to fit this nested $Q$-value function and allow for group imbalance in the data.

3. We show that NFQI generates policies that outperform baseline methods on a set of quantitative and qualitative metrics and rely on relevant state features to do so.

This paper proceeds as follows. Section 2 discusses related work. In Section 3, we present a framework for NFQI. In Section 4, we show results for NFQI in simulations and hospital data. Section 5 concludes and discusses potential future work.

## 2 RELATED WORK

**Multi-task RL.** The goal of multi-task RL is to train an agent to perform optimally at multiple tasks (8). Recent work has developed a method to efficiently learn a task-conditioned policy $\pi(a|s, z)$, where $a$ is an action, $s$ is a state, and $z$ is a task embedding (30). This approach has been observed to perform better than single-task learning because it uses the relationships between tasks to improve performance for individual tasks (27). However, the goals of multi-task RL differ from those of NFQI: Multi-task RL optimizes performance for multiple tasks, each with a different Markov decision process (MDP), whereas NFQI is most appropriate to compare two environments with identical tasks and reward functions but different transition dynamics. Additionally, the neural networks typically used for multi-task RL (30) require large training sets that are not always available in the medical and learning applications NFQI was built to address. As such, these networks are likely to suffer from overparameterization in these applications.

**Meta RL.** Meta RL algorithms train agents to solve new tasks efficiently based on experience with other similar tasks. These approaches infer the task from a distribution $p(T)$ using a small set of observations, rather than through an explicit task embedding. Much of the effort in meta RL is focused on task design as opposed to model tuning, to make this inference process easier (12). Other work has shown that meta RL can be appropriate for offline RL (21). Like multi-task RL, meta RL is different from NFQI because it trains agents to perform multiple tasks with different state and action spaces. Unlike meta RL and multi-task RL, NFQI expects all samples to arise from MDPs that have the same state and action spaces and follow the same reward function. Since the dynamics of the two environments vary, the MDPs will have different state transition functions, entailing different optimal policies for the groups.

**Transfer learning.** Transfer learning methods fine-tune a pre-trained model for a target dataset. Often, the target dataset is relatively small; transfer learning using a larger and similar dataset may result in better performance than training a model on just the target dataset (6). A common way to use these methods is to first train a neural network using the large dataset; then, some of the network layers are frozen, and the rest of the network is fine-tuned by training on the target dataset. In this way, the network is adapted to the target problem by learning a nonlinear map from the solution estimated for the larger dataset to the outcome for the target dataset. This strategy can be applied to reinforcement learning (31) and may improve resource usage when the target dataset is small (4). The two datasets that NFQI uses are likely to have substantial sample size imbalance; for this reason, we train the parameters for NFQI using a procedure inspired by transfer learning.

## 3 METHODS

In this section we present nested policy fitted Q-iteration (NFQI), an off-policy RL algorithm for environments that have a nested group structure. We first review $Q$-learning and fitted Q-iteration (FQI) and then describe how to extend them to NFQI.

### 3.1 NOTATION AND PRELIMINARIES

A Markov decision process (MDP) is a discrete-time stochastic control process defined by a 4-tuple $(\mathcal{S}, \mathcal{A}, P, R)$, where $\mathcal{S}$ is the state space; $\mathcal{A}$ is the set of actions; $P(\mathbf{s}_{t+1}|\mathbf{s}_t, \mathbf{a}_t)$ is a function

governing state transition probabilities from time $t$ to $t+1$; and $R : \mathcal{S} \to \mathbb{R}$ is a reward function. We assume a finite number of actions. A policy $\pi : \mathcal{S} \to \mathcal{A}$ is a mapping from states to actions that describes an agent's strategy within an MDP. In general, we assume $\mathbf{s}_t \in \mathbb{R}^p$ and $\mathbf{a}_t \in \mathbb{R}^q$.

## 3.2 $Q$-LEARNING AND FITTED $Q$-ITERATION (FQI)

Fitted Q-iteration (FQI) is an off-policy RL algorithm based on $Q$-learning (9; 29) that is designed to find an optimal policy for a given MDP. $Q$-learning approaches define a function $Q : \mathcal{S} \times \mathcal{A} \to \mathbb{R}$ that maps state-action pairs to an estimate of the cumulative expected reward obtained by taking a particular action in a particular state. For a given policy $\pi$ at time step $t$, $Q$ is defined as $Q^\pi(s_t, a_t) = r_t + \gamma \mathbb{E}_{\mathbf{s}'}[V^\pi(\mathbf{s}')]$, where the expectation is taken over the possible next states given the transition probabilities $P$, $V^\pi$ is the value function corresponding to $\pi$, and $\gamma \in [0,1]$ is a discount factor. The principle of Bellman optimality (5) states that a policy $\pi^\star$ is optimal for a given MDP if and only if, for every state $\mathbf{s} \in \mathcal{S}$, it holds that $V^{\pi*}(\mathbf{s}) = \max_{\mathbf{a}} R(\mathbf{s}, \mathbf{a}) + \gamma \mathbb{E}_{\mathbf{s}'}[V^{\pi*}(\mathbf{s}')]$. Broadly, $Q$-learning algorithms attempt to approximate the $Q$ function using agents' observed interactions with an environment, as the state transition function is typically not directly observable. This approximation is used to recommend optimal actions for a given state. The data used to estimate the $Q$ function typically consist of samples representing steps taken within an MDP. In particular, a sample at time $t$ is a 4-tuple $(\mathbf{s}_t, \mathbf{a}_t, \mathbf{s}_{t+1}, r_t)$ representing the agent's state, chosen action, observed next state, and observed reward.

FQI approximates $Q$ using an offline batch optimization approach (9). Specifically, it first posits a family of functions $f_\theta(\mathbf{s}, \mathbf{a})$, parameterized by $\theta$, that are meant to approximate the expected future reward after taking action $\mathbf{a}$ from state $\mathbf{s}$. Common choices for $f$ are regression trees (9) and neural networks (25). FQI then employs an iterative algorithm that alternates between updating the current $Q$-values and fitting a regression problem that optimizes the parameters $\theta$. Suppose we have a set of $T$ samples from an MDP, represented as a set of one-step transition tuples $\{(\mathbf{s}_t, \mathbf{a}_t, \mathbf{s}_{t+1}, r_t)\}_{t=1}^T$. On step $k$ of the algorithm, FQI first updates its estimate of the $Q$-function according to the Bellman equation: $\widehat{Q}_k(\mathbf{s}_t, \mathbf{a}_t) = r_{t+1} + \gamma \max_{\mathbf{a}' \in \mathcal{A}} f_{\theta_{k-1}}(\mathbf{s}_{t+1}, a')$ for $t = 1, \ldots, T$. The second step minimizes a loss measuring the discrepancy between the current $Q$-function estimate and the approximating function $f$. The optimization problem is $\min_{\theta_k} \sum_{n=1}^N \mathcal{L}\left(\widehat{Q}_k(\mathbf{s}_t^n, \mathbf{a}_t^n), f_{\theta_k}(\mathbf{s}_t^n, \mathbf{a}_t^n)\right)$, where $\mathcal{L}(y, \widehat{y})$ is the loss function comparing the "true" value $y$ to predicted value $\widehat{y}$. After $K$ iterations, FQI's approximation to the optimal policy is given by $\pi(\mathbf{s}) = \arg\max_{\mathbf{a}' \in \mathcal{A}} f_{\theta_K}(\mathbf{s}, \mathbf{a}')$.

## 3.3 NESTED POLICY FQI

We now introduce NFQI, an extension of FQI that estimates group-specific policies and accounts for a nested group structure in the data. NFQI is a framework for off-policy reinforcement learning that can use an arbitrary function class to approximate the $Q$ function. Below, we describe the general NFQI framework with minimal assumptions about its parameterization. Additional experiments show that neural networks are well suited to parameterize $Q$ (see Appendix 6.2 and Appendix 6.5).

Recall that nested datasets are structured with two predefined groups – the background, and foreground. To account for nested datasets, NFQI imposes structure on the family of functions used to approximate the $Q$ function. We define the approximating function $f$ as

$$f(\mathbf{s}, \mathbf{a}, z) = g_s(\mathbf{s}, \mathbf{a}) + \mathbb{1}_{\{z=1\}} g_f(\mathbf{s}, \mathbf{a}), \tag{1}$$

where $g_s$ is a function modeling the shared $Q$-value structure between the background and foreground samples, $g_f$ models foreground-specific structures, and $\mathbb{1}_{\{z=1\}}$ is an indicator function that returns 1 when $z = 1$ (foreground) and 0 otherwise (background). Note that, despite the unconventional structure of the approximating function, it is still guaranteed to converge to the Bellman optimal $Q$-value function; see Appendix 6.1 for further details.

NFQI uses a two-stage training procedure inspired by transfer learning (Appendix Algorithm 1). In the first stage, we train $g_s$ using all of the foreground and background training samples, while ignoring the foreground-specific model $g_f$. Note that this stage is equivalent to setting $z = 0$ for all samples (including foreground samples) temporarily. In the second stage, we train the foreground-specific function $g_f$ using only foreground training samples. Under this approach, we first estimate the shared structure of the network using both groups of samples in our training set, and we fine-tune the network's foreground predictions in $g_f$ using just the foreground training samples.

## 4 EXPERIMENTS

We demonstrate the performance of NFQI through the following experiments. Our experiments seek to answer the following questions:

1. How well does a joint policy learned using NFQI perform in comparison to two policies learned independently (one for each dataset) using FQI?

2. How does a joint policy learned using NFQI perform compared to a joint policy learned by applying FQI or transfer learning to the union of background and foreground datasets?

3. Do the policies learned by NFQI rely on group-specific features to make predictions?

4. Are policies learned using NFQI able to achieve good performance when the sample size of the background dataset is much larger than that of the foreground dataset?

5. When there is no group structure in the dataset and the application of NFQI is technically inappropriate, does NFQI perform as well as standard FQI?

We present results from NFQI applied to a nested Cartpole environment (7) and to a clinical decision support task using MIMIC-IV, a benchmark electronic health record (EHR) dataset (14; 11).

### 4.1 NESTED CARTPOLE

We first demonstrate the performance of NFQI using the Cartpole environment, as implemented in the OpenAI gym (7). The Cartpole environment consists of an unstable pole attached to the top of a cart. An agent's objective is to keep the pole balanced upright (in this case for 1000 steps); this can be done by strategically applying a force on the cart to the left or right at each time step (see Appendix 6.4 for the full MDPs and Appenxix 6.6 for training details).

In this study, we adapt the Cartpole environment to accommodate a nested structure. We call our setup the "nested Cartpole" environment. The background environment is the original Cartpole setup, while the foreground environment includes a constant force of $c$ Newtons that pushes the cart to the left. Unless otherwise specified, $c = 5$. Although the objectives for the two simulators is identical, an agent will require a different policy to succeed in the background and foreground environments because the dynamics of each system are different. Intuitively, an agent in the foreground environment will need to develop a policy that counteracts the consistent leftward force.

To construct datasets we sample Cartpole episodes from the foreground and background environments. Each trajectory starts in a random state and chooses actions uniformly at random until the pole falls over. We use these trajectories to train NFQI and other off-policy algorithms. We evaluated the performance of each policy by applying it in the nested Cartpole simulator and computing the number of steps the cart maintains the pole in the goal state (with a maximum of 1000 steps).

#### 4.1.1 NFQI IMPROVES PERFORMANCE WHEN THE FOREGROUND SAMPLE SIZE IS SMALL

We first sought to benchmark NFQI's performance against FQI. To do so, we generated a training set of trajectories that reflect the group imbalance that we expect to see in the nested setting. In particular, this dataset contains 200 background samples and 50 foreground samples. The foreground environment has a force of $c = 5$ Newtons pushing the cart to the left. We trained FQI separately on the background and foreground dataset; this yielded two independent policies. We fit NFQI with both datasets using the procedure in Appendix Algorithm 1. We tailored the neural networks for each algorithm so that they both use the same number of parameters (see Appendix 6.6), thus precluding either algorithm from having a more computationally expressive family of functions. We evaluated each policy 50 times in each of the foreground and background environments, computing the mean and confidence interval of these repetitions.

We find that the policy learned by NFQI performs substantially better than FQI in the foreground environment and slightly better than FQI in the background environment (Fig. 1). FQI's poor performance in the foreground environment can likely be explained by the small training set size. However, NFQI is able to leverage information from both foreground and background trajectories, which allows it to overcome this limited sample size. This result suggests that NFQI is a viable

algorithm for learning optimal policies for two groups, especially when the training set sizes are unbalanced between the groups.

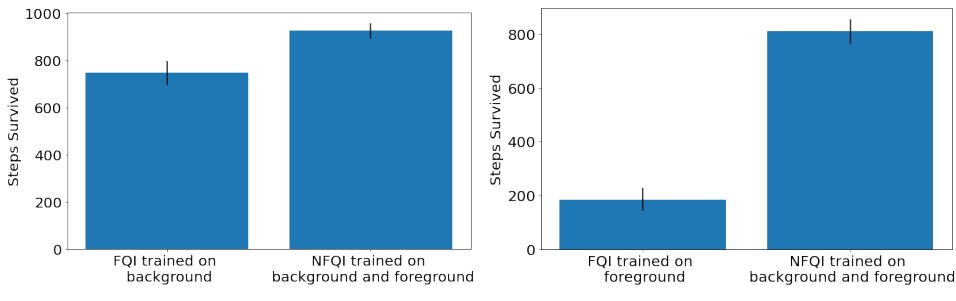

(a) Background environment performance   (b) Foreground environment performance

Figure 1: Performance of NFQI and FQI in the background (Panel a) and foreground (Panel b) environments. The bar height represents the mean number of steps in the goal state (with a maximum of 1000 steps); ticks represent 95% confidence intervals over 50 repetitions. NFQI trained with both datasets performs better on test data from each of the environments, and substantially better for the foreground test data.

### 4.1.2 NFQI OUTPERFORMS FQI AND TRANSFER LEARNING

Given that we want to train a policy using both datasets at the same time, we next compared NFQI in the nested Cartpole environment to two other bespoke approaches: FQI trained jointly with trajectories generated from both the foreground and background environments, and transfer learning. The FQI and transfer learning algorithms find a single policy that does not consider a group label at training or inference time (further details in Appendix 6.6).

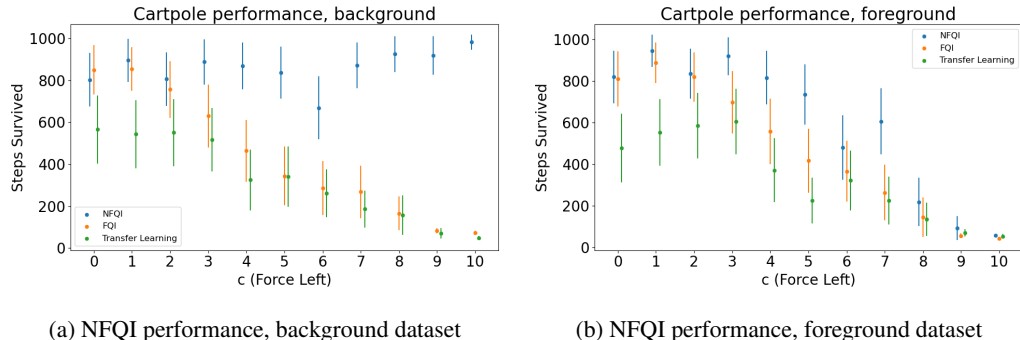

(a) NFQI performance, background dataset   (b) NFQI performance, foreground dataset

Figure 2: NFQI outperforms related algorithms in a nested Cartpole environment. Increasing the force on the cart (x-axis) increases the difference in the foreground and background environments. As the environments become more distinct, NFQI enables agents to survive longer (y-axis) than FQI and transfer learning for both the background (Panel a) and foreground (Panel b) data.

To evaluate the algorithms, we created eleven variations of the nested Cartpole environment, each corresponding to a different value for the constant force applied in the foreground, $c \in \{0, 1, \ldots, 10\}$. We evaluate each algorithm 15 times for each environment.

We found that NFQI outperforms FQI and transfer learning in this setting, especially as the foreground and background environments become more different from one another (Fig. 2). When the foreground and background environments were the same (i.e., $c = 0$), NFQI and FQI performed similarly, while transfer learning performed slightly worse. As the background and foreground dynamics became more distinct (i.e., as $c$ increased), NFQI outperforms FQI. In particular, NFQI maintained near-perfect performance in the background, and the performance degraded more slowly in the foreground environment compared to competing approaches. These results suggest that NFQI

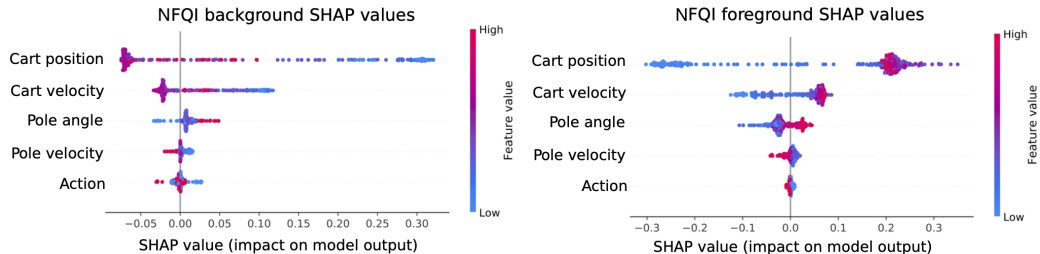

Figure 3: SHAP plots for background (Panel a) and foreground (Panel b) samples from the Cartpole environment. Each point represents the SHAP value for a single sample and single feature. The magnitude of SHAP values is similar in both contexts, but foreground SHAP values are shifted right, suggesting that the foreground policy is compensating for the foreground-specific dynamics.

is able to model the differences between the background and foreground while exploiting shared structure to improve its performance in both environments.

### 4.1.3 NFQI YIELDS POLICIES THAT RELY ON RELEVANT FEATURES

In addition to obtaining high-performance policies, it is also desirable to have these policies rely on relevant features. To investigate this, we computed estimates of feature importance for both the shared and foreground-specific model components using SHapley Additive exPlanations (SHAP), a game theoretic approach to quantify the extent to which the features of each sample impact the model's predictions (16). Features with larger SHAP absolute values indicate a greater predictive power. In the nested Cartpole environment, we compute SHAP values for each of the five features used to estimate $Q$-values: *cart position, cart velocity, pole angle, pole velocity*, and *action*.

We observed several patterns in the SHAP values for NFQI that agree with prior intuition. First, we found that across all samples in both datasets, the features rank in the same order of importance. This ordering corresponds to the mean SHAP value across samples. We see that *cart position* shows the highest SHAP values, meaning it is the most predictive feature, while *action* is the least predictive (Fig. 3). The model's emphasis on *cart position* is likely due to the constant force being applied in the foreground Cartpole environment; NFQI's policy adapts to the relative importance of this feature for both environments. We see that the foreground *cart position* SHAP magnitude is much larger relative to that in the background, indicating that the foreground policy is compensating for the foreground-specific dynamics, which include a constant force pushing the cart towards the left. These results suggest that the policies found by NFQI rely on relevant features, as demonstrated in part by the SHAP values.

### 4.1.4 NFQI IS ROBUST TO GROUP SIZE IMBALANCE

In many real-world settings, we have access to far less data in the foreground condition than in the background. For example, it is much easier to collect medical records from healthy (background) patients than from patients with a specific, chronic disease (foreground). Thus, it is important that nested RL methods be resilient to group imbalance and able to leverage the statistical strength of the background data to estimate foreground-specific structure even when foreground data are rare. In order to test NFQI's performance with imbalanced data, we conducted an experiment with imbalanced group sizes in the nested Cartpole environment. Similar to previous experiments, we gave the foreground different dynamics than the background by adding a constant leftward force of $c = 5$ Newtons. Here, we fix the total number of training trajectories across both conditions to be $400$, and we set the fraction of samples that come from the foreground data to $10\%, 20\%, 30\%, 40\%$, and $50\%$ (where $50\%$ corresponds to balanced sizes). We choose these fractions because they reflect percentages of imbalance that we see in medical data. We fit NFQI as before, using a neural network as our $Q$-value approximation function. For comparison, we also train FQI on the combined dataset of background and foreground trajectories. We repeat this experiment $50$ times.

NFQI outperforms FQI in this Cartpole setting with imbalanced group sizes. Specifically, across the range of data imbalance, the policies found by NFQI perform better than the FQI policies in both

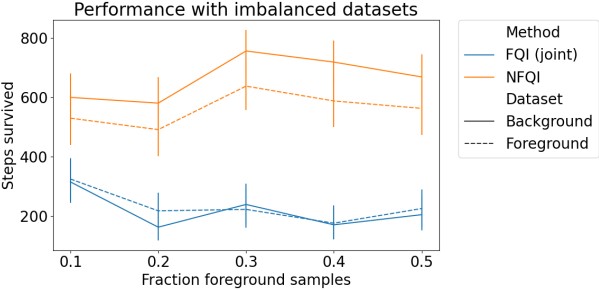

Figure 4: NFQI is robust to imbalance in foreground and background sample sizes. We fix the total number of training trajectories across both conditions to be 400, and we set the fraction of samples that come from the foreground data to $10\%, 20\%, 30\%, 40\%$, and $50\%$ (where $50\%$ corresponds to balanced sizes). Error bars correspond to $95\%$ confidence intervals. NFQI maintains superior performance to FQI despite increasing levels of class imbalance. The foreground dataset is harder to model than the background, but a nested policy better reflects the environments.

the background and foreground (Fig. 4). This is likely due to the fact that NFQI accounts for the group structure in the dataset, while also exploiting the shared structure between the background and foreground to increase the effective sample size. On the other hand, FQI must model the groups with a single joint model, resulting in poor performance for both datasets. This result implies that NFQI is successful in a common setting in which we have access to a small proportion of foreground samples relative to the background samples.

### 4.1.5 NFQI ONLY LEARNS DIFFERENT POLICIES IF THE ENVIRONMENTS ARE DIFFERENT

When there is no meaningful group structure in the data, we want NFQI to find the same optimal policies for two (identical) groups. We demonstrate that NFQI behaves in this way by testing its performance when there is no nested structure. We call this experiment the "structureless" test. We first generate a dataset from a single simulator and give it false group structure by randomly assigning group labels to samples. Then, we fit NFQI to this dataset and analyzed the policies for each group, again using SHAP plots.

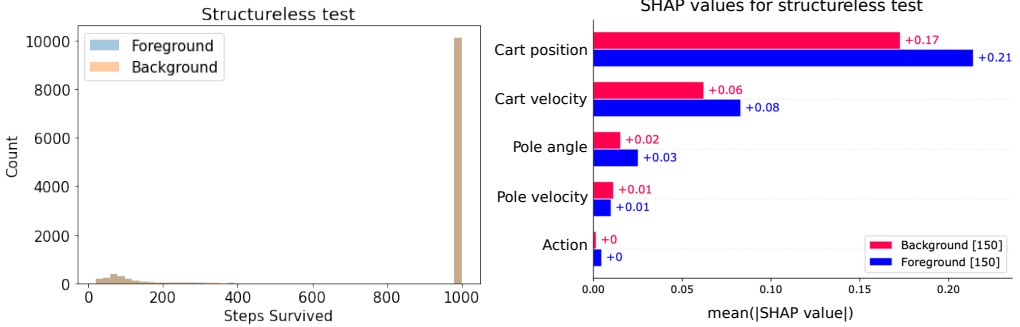

(a) Structureless test for the Cartpole environment.      (b) SHAP plot for the structureless test.

Figure 5: NFQI does not estimate practically different policies for two groups when there is no group structure. Here, we find that NFQI finds policies that yield similar performance for both the foreground and background group (Panel a) and that NFQI uses sample features similarly across groups to predict rewards (Panel b).

We find that NFQI finds two policies that show similar behavior between the artificial background and foreground datasets (Fig. 5a). The policies use the sample features similarly to make predictions, with the same rankings of mean absolute SHAP values across artificial groups (Fig. 5b). These results suggest that NFQI learns indistinguishable policies when the samples originate from the same simulator with identical dynamics.

## 4.2 CLINICAL DECISION MAKING TASK USING ELECTRONIC HEALTH RECORDS (EHR)

After validating NFQI on the Cartpole environments, we next set out to test NFQI's ability to identify more complex policies for patient treatment in a hospital setting. To do so, we leveraged electronic health record (EHR) data from the Medical Information Mart for Intensive Care (MIMIC) series (22; 26; 15; 14), which contains de-identified records from patient hospital admissions.

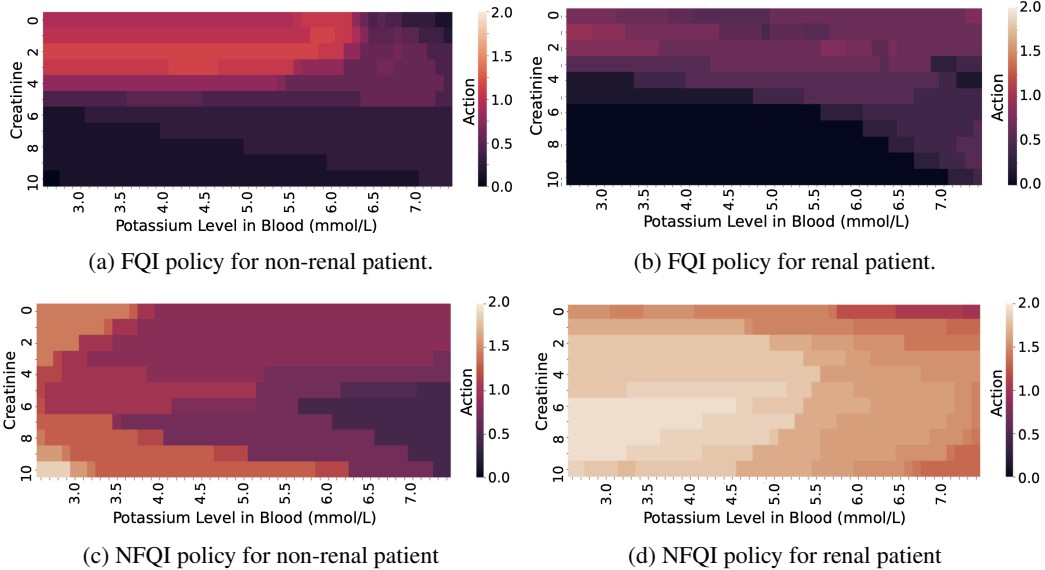

(a) FQI policy for non-renal patient.

(b) FQI policy for renal patient.

(c) NFQI policy for non-renal patient

(d) NFQI policy for renal patient

Figure 6: Visualizing FQI and NFQI policies for non-renal and renal patients. Heatmaps indicate the policy's repletion recommendation for patients with a given potassium and creatinine level.

In this experiment, we aim to identify a policy for maintaining healthy levels of electrolytes in hospital patients. Determining when to prescribe – or "replete" – electrolytes is a difficult task; the overuse of medication intended to balance electrolyte levels in patients has been linked to patient mortality and complications post-surgery (28). Here, we use NFQI to identify electrolyte repletion policies for two groups of intensive care unit (ICU) patients that require different repletion strategies: a foreground group of 123 patients who have kidney disease (renal patients) and a background group of 400 patients with healthy kidney function (non-renal patients). Our goal is to determine the dosage of potassium to administer to a patient across the duration of their hospital stay. We discretize the action space into three levels of repletion: *no repletion*, *low repletion*, and *high repletion*.

Table 1: **Evaluating RL algorithms using action matching.** We evaluate FQI and NFQI on their ability to predict the clinician's actions on test samples from the foreground (FG) and background (BG) datasets. We report accuracy (%) and F1 score across the three classes of repletion, both with 95 % confidence intervals.

| Algorithm | FG Accuracy (%) | FG F1 Score | BG Accuracy (%) | BG F1 Score |
|---|---|---|---|---|
| FQI | $36.09 \pm 8.5$ | $\mathbf{0.26 \pm 0.05}$ | $27.7 \pm 6.6$ | $0.20 \pm 0.03$ |
| NFQI | $\mathbf{45.5 \pm 12.2}$ | $0.25 \pm 0.04$ | $\mathbf{35.7 \pm 6.3}$ | $\mathbf{0.26 \pm 0.03}$ |

Building off previous RL approaches for this task (24), we discretize patient trajectories into 6-hour intervals. Each sample contains information about the patient's vitals, labs, and prescribed medicines during that interval, as well as comorbidities and metadata, such as age. The reward function for this dataset is a manually parameterized function of the patient state and repletion action that produces higher values when a patient's measured potassium level is within a healthy range (3.5 and 4.5 mmol/L), and lower reward values otherwise (24). This reward function must be constructed manually because there is no notion of reward in EHR records, and we cannot derive an empirical one because it is not safe or ethical to evaluate a policy on patients (see Appendix 6.7 for further details).

We fit NFQI and FQI to this dataset and examine the resulting policies, performing both quantitative and qualitative analyses on our policies. We quantitatively evaluate a policy's performance based on its level of agreement with a doctor's actions in test data (Tab. 1). We find that the policies estimated using NFQI better reflect doctor's actions than those estimated with FQI.

To give us better insight into why NFQI quantitatively outperforms FQI, we visualize the policies qualitatively through heatmaps. We create synthetic patient states by densely sampling a grid with two features, potassium and creatinine, while holding all other state features fixed. Creatinine levels capture the kidney's ability to filter out toxins (18); higher levels of creatinine indicate worse kidney function. Given a patient's state features, we then use the trained $Q$-value function to select the action that generates the highest expected reward. NFQI recommends higher levels of repletion for renal patients versus non-renal patients, especially when the patient's potassium is below the healthy range (19) (Fig. 6). Moreover, NFQI's policy recommends more repletion for renal patients with higher creatinine level, indicating that electrolyte repletion should increase as kidney function worsens. Meanwhile, FQI's policy shows little difference between these two patient groups, and recommends lower levels of repletion across the grid. These results suggest that NFQI is able to discriminate between the patient groups and act according to prior intuition.

It is also imperative that NFQI policies are interpretable since the actions recommended can affect other aspects of a patient's physiology. Thus, we studied which features NFQI found most predictive in the foreground and background policies. To do so, we computed SHAP values from each group (Fig. 7). We find that NFQI identifies clinically relevant characteristics to inform both foreground and background policies. Patients with renal disease tend to have higher creatinine values and fluctuating electrolyte levels; the foreground policy for renal patients relies on potassium and creatinine measurements more than the background policy. This suggests that the NFQI policies reflect the group-specific dynamics, while also sharing strength across the foreground and background samples.

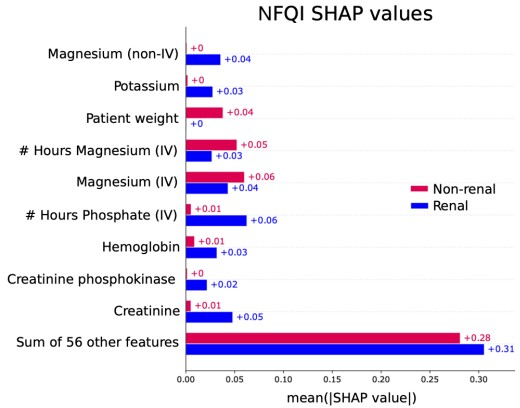

Figure 7: NFQI mean SHAP values for renal (blue) and non-renal (red) patients. Y-axis shows clinically relevant covariates.

## 5 DISCUSSION AND FUTURE WORK

In this work, we present nested policy fitted Q-iteration (NFQI), an off-policy RL framework that estimates optimal policies for nested environments. We demonstrate that NFQI policies outperform those estimated from FQI and related approaches. When applied to simulated and medical data, these policies are performant, robust to sample size imbalance, learn justifiable differences in the datasets, and gracefully revert to FQI policies when there is no relevant group structure.

Several future directions remain. From a methods perspective, other approaches for the nested $Q$-value function could be considered, such as Gaussian processes, random forests, and other neural network architectures. This may improve the fit to new, larger datasets by increasing the expressiveness of the $Q$-value approximation function. Additionally, extending the nested model to allow for multi-group structure may be useful when there are more than two groups in the data. This may involve changing the structure of the approximating function $f$ to incorporate other compositions of $g_s$ and $g_f$. In medical settings we anticipate future work in enabling NFQI as a clinical decision-support tool (i.e., to recommend treatments with qualitative justification) to be used in conjunction with a physician's expertise.

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

## 6 APPENDIX

### 6.1 BELLMAN OPTIMALITY OF NFQI'S $Q$-VALUE FUNCTION

In its original formulation, FQI was shown to converge to a Bellman-optimal $Q$-value function using concepts from dynamic programming theory (10). Here, we show that NFQI reduces to a particular type of FQI, and thus remains within this Bellman-optimal family.

Recall that FQI approximates the $Q$-value function with a function $f_{\text{FQI}}(\mathbf{s}, \mathbf{a})$ taking a state-action pair as input, while NFQI uses an approximating function $f_{\text{NFQI}}(\mathbf{s}, \mathbf{a}, z)$ that accepts an additional argument $z$ representing the sample's group identity. Because $z$ is an attribute of the environment, we can think of it as a dimension of the state space that takes two possible values, and is constant over a given trajectory. Thus, trivially we can augment the state as $\widetilde{\mathbf{s}} = (\mathbf{s}^\top, z)^\top$ and describe NFQI as a particular case of FQI where the approximating function is $f_{\text{FQI}}(\widetilde{\mathbf{s}}, a)$. We can write the additive form of NFQI in Equation 1 as

$$f_{\text{FQI}}(\widetilde{\mathbf{s}}, \mathbf{a}) = g_s(\mathbf{s}, \mathbf{a}) + \mathbb{1}_{\{\widetilde{\mathbf{s}}_{p+1}=1\}} g_f(\mathbf{s}, \mathbf{a})$$

where $\mathbf{s}$ is a $p$-vector, so the $(p+1)$th element of $\widetilde{\mathbf{s}}_{p+1}$ is $z$.

## 6.2 Neural network Q-value functions

Here, we present one instance of NFQI using a neural network as a nonlinear $Q$-value function approximator.

For simplicity, we describe an implementation using a multilayer perceptron (MLP) with one hidden layer; the nested network can be generalized to arbitrarily large architectures. Let $\sigma$ be an element-wise nonlinear activation function, and suppose that each state-action pair is $p$-dimensional, which we write as a single vector $\mathbf{x}_t = [\mathbf{s}_t^\top, \mathbf{a}_t^\top]^\top \in \mathbb{R}^p$. Then, the output of the network for a background sample is given by

$$f(\mathbf{s}_t, \mathbf{a}_t, 0) = \underbrace{\sigma(\mathbf{x}_t^\top \mathbf{W}_{1s})\mathbf{w}_{2s}}_{g_s(\mathbf{s}_t, \mathbf{a}_t)},$$

where $\mathbf{W}_1 \in \mathbb{R}^{p \times k}$ and $\mathbf{w}_2 \in \mathbb{R}^k$ are linear network weights separated by the nonlinear function $\sigma$. Note that, because this is a background sample, we only apply $g_s$ to the sample. To model foreground samples, we also include a foreground-specific additive component in the network. This component, $g_f$, only operates on foreground samples. The output of the network for a foreground sample is given by:

$$f(\mathbf{s}_t, \mathbf{a}_t, 1) = \underbrace{\sigma(\mathbf{x}_t^\top \mathbf{W}_{1s})\mathbf{w}_{2s}}_{g_s(\mathbf{s}, \mathbf{a})} + \underbrace{\sigma(\mathbf{x}_t^\top \mathbf{W}_{1f})\mathbf{w}_{2f}}_{g_f(\mathbf{s}_t, \mathbf{a}_t)},$$

where $\mathbf{W}_{\ell s}$ and $\mathbf{W}_{\ell f}$ are the shared and foreground-specific weights for layer $\ell \in \{1, 2\}$, respectively.

In our experiments below, we use a variant of this type of neural network (6.6). We optimize the network parameters $\theta = \{\mathbf{W}_{\ell s}, \mathbf{W}_{\ell f}\}_{\ell=1}^L$ using a squared error loss function,

$$\min_\theta \sum_{n=1}^N \left( \widehat{Q}(\mathbf{s}_t, \mathbf{a}_t) - f_\theta(\mathbf{s}_t, \mathbf{a}_t, z_t) \right)^2.$$

## 6.3 Code

Before NFQI can be deployed in a particular application area, its hyperparameters, and possibly its model structure, should be tuned. We release code with recommendations for tuning these settings. All of our experiments were run on an internally-hosted cluster using a 320 NVIDIA P100 GPU whose processor core has 16 GB of memory hosted. Our experiments used a total of approximately 100 hours of compute time. The code for our experiments is based on a prior implementation of FQI[1]. We implement our nested neural network models in PyTorch (23) and use an SGD-based optimizer (1). For all experiments, we use 80% of our data to train and 20% of our data to test. We use a default learning rate of $10^{-3}$. We use the same hyperparameters for nested- and group-label agnostic methods.

## 6.4 Cartpole environment

The Cartpole environment contains a cart with an inverted pendulum pole on top, and an agent's goal in this environment is to move the cart left and right such that the pole remains upright and the cart itself stays within the bounds of the window box. More formally, we can describe the Cartpole environment as a Markov decision process (MDP) represented by a tuple $(\mathcal{S}, \mathcal{A}, P, R)$. Here, $\mathcal{S} = [x, \dot{x}, \theta, \dot{\theta}]^\top$ is a four-dimensional state space, where $x$ and $\dot{x}$ are the cart's position and velocity, and $\theta$ and $\dot{\theta}$ are the pole's angle and angular velocity; $\mathcal{A} = \{a_\ell, a_r\}$ is the action space, containing actions for applying a force of 10 Newtons to the left or right; $P$ is a deterministic function determining state dynamics (based on simple Newtonian mechanics in this case); and $R$ is a reward function, which returns $-1$ when the pole falls or the cart exits the frame (at which point the episode is terminated), and returns 0 otherwise. In the *nested* Cartpole environment, we adjust the dynamics described by $P$ to simulate different dynamics for the foreground and background environments. Specifically, in the foreground environment, we include a constant leftward force of $c$ Newtons. In this case, choosing the action "push left" results in a leftward force of $10 +$

---
[1]https://github.com/seungjaeryanlee/implementations-nfq

$c$ Newtons, while the action "push right" results in a rightward force of $10 - c$ Newtons. This asymmetry is the driving difference between the background and foreground environments. As such, each environment has the same state space, action space, and reward function, but a different state transition function.

## 6.5 LINEAR MODEL $Q$-VALUE FUNCTION

We tested the effect of the choice of the approximating function $f$ on the performance of the NFQI framework. To do so, we fit NFQI and FQI using a linear model and a neural network in the Cartpole environment.

For the linear model, we consider $g_s$ and $g_f$ to be linear models in Equation equation 1. We represent each sample by a vector containing its state and action, $[\mathbf{s}_t^\top, \mathbf{a}_t^\top]^\top$, where $\mathbf{s}_t$ is the state vector and $\mathbf{a}_t$ is the action vector. We assume a finite action space throughout this study. Then, we have the following model:

$$g_s(\mathbf{s}_t, \mathbf{a}_t) = \begin{bmatrix} \mathbf{s}_t \\ \mathbf{a}_t \end{bmatrix}^\top \boldsymbol{\beta}_s + \mathbf{1}^\top \boldsymbol{\beta}_{0s} + \epsilon$$

$$g_f(\mathbf{s}_t, \mathbf{a}_t) = \begin{bmatrix} \mathbf{s}_t \\ \mathbf{a}_t \end{bmatrix}^\top \boldsymbol{\beta}_f + \begin{bmatrix} \mathbf{s}_t \\ \mathbf{a}_t \end{bmatrix}^\top \boldsymbol{\beta}_s + \mathbf{1}^\top \boldsymbol{\beta}_{0f} + \epsilon,$$

where $\mathbf{1}$ represents a column vector of ones, and $\epsilon$ is zero-mean random noise. Here, the coefficients $\beta_s$ and $\beta_f$ capture the background and foreground's (linear) relationship between state and action pairs, and expected reward. The terms $\boldsymbol{\beta}_{0s}$ and $\boldsymbol{\beta}_{0f}$ capture context-specific intercepts. Note that this approach is similar to a hierarchical model approach (20; 3).

As expected, we find that the linear model is unable to successfully find a policy that solves the Cartpole environment (Figure 8). In contrast, a neural network is able to find a good policy most of the time. We anticipate this is because a neural network can approximate functions that are nonlinear and that the Cartpole environment requires a nonlinear function to approximate reward. We also expect the MIMIC-IV environment to require a nonlinear function. For this reason, we consider neural networks throughout this study.

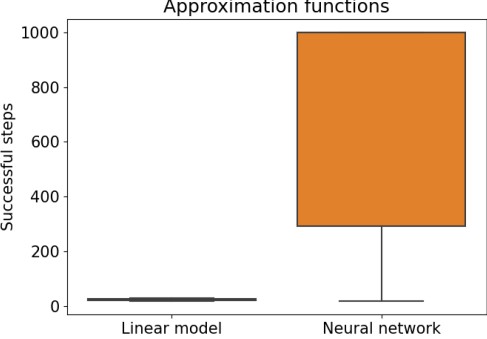

Figure 8: Performance of FQI in the Cartpole environment using two different approximation functions. Performance of each model is measured as the number of steps before the pole fell or the cart exited the frame. Performance is shown here for a linear model and a neural network.

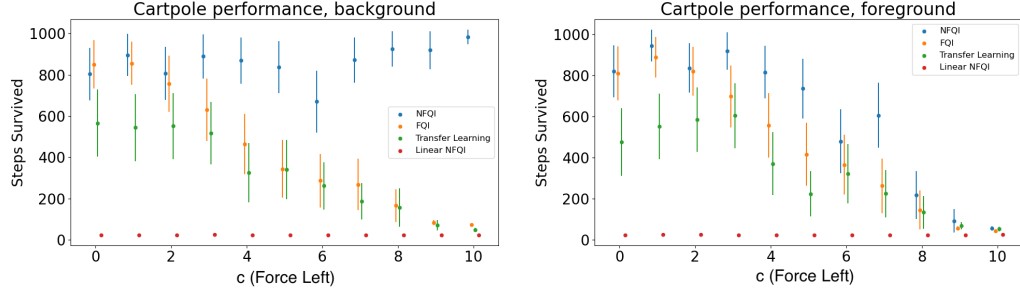

Figure 9: A neural network-based version of NFQI outperforms related algorithms and a linear model-based NFQI in a nested Cartpole environment. Linear NFQI is likely underparameterized for the Cartpole problem in both the background (Panel a) and foreground (Panel b) environments.

---

**Algorithm 1:** NFQI training procedure

---

Randomly initialize shared and foreground-specific model parameters $\theta_s, \theta_f$;
**while** *background loss $\mathcal{L}_b$ not converged* **do**
    | Compute $\mathcal{L}_b$;
    | Update $\theta_s$ using SGD;
**end**
**while** *foreground loss $\mathcal{L}_f$ not converged* **do**
    | Compute $\mathcal{L}_f$;
    | Update $\theta_f$ using SGD;
**end**
$\pi_b(\mathbf{s}) \leftarrow \arg\max_{\mathbf{a}'} g_s(\mathbf{s}, \mathbf{a}')$ `// background policy`
$\pi_f(\mathbf{s}) \leftarrow \arg\max_{\mathbf{a}'} g_f(\mathbf{s}, \mathbf{a}')$ `// foreground policy`
**return** $\pi_b, \pi_f$

---

## 6.6 ALGORITHMS

We use three algorithms in our experiments: NFQI, FQI, and transfer learning. We use the same network with the same number of parameters for all three algorithms. The difference between the algorithms can be explained using the training procedures as described below. The network structure follows:

$$\text{input} = \text{s, a, z}$$
$$\text{x} = [\text{s, a}] \longrightarrow \text{Linear(10)} \longrightarrow \text{ReLU} \longrightarrow \text{Linear(5)} \longrightarrow \text{ReLU}$$
$$\text{x\_shared} = \text{x} \longrightarrow \text{Linear(1)}$$
$$\text{x\_fg} = \text{x} \longrightarrow \text{Linear(10)} \longrightarrow \text{ReLU} \longrightarrow \text{Linear(5)} \longrightarrow \text{ReLU} \longrightarrow \text{Linear(1)}$$
$$\text{output} = \text{x\_shared} + \text{z} \times \text{x\_fg}$$

With the exception of the imbalanced dataset experiment (Section 4.1.4), all algorithms use the same number of training and test samples; the network is trained using only training samples, and tested using only test samples.

To train NFQI, we follow Algorithm 1. To perform inference, we input a state-action pair and the group label to the network. Depending on the group label, the network uses either the shared layers or both the shared and foreground layers to estimate a $Q$-value. To train FQI, we do not consider the group label of the sample. The network uses all of the layers to estimate a $Q$-value for each training sample. This is as if all of the group labels were 1.

We also do not consider the group label when training a transfer learning algorithm. We first use the background training samples to train the shared layers in our network. Then, we freeze the shared layers and use the foreground training samples to train the foreground specific layers. When performing inference, this network does not consider group label; as such, the network uses every layer to estimate a $Q$-value for a sample.

All of our algorithms use a mini-batch SGD optimizer and an MSE loss. A random seed is used to ensure reproducibility. We evaluate the network on one test sample for every train sample in the batch; if the previous three evaluations were successful (i.e., the pole in the Cartpole environment stayed up for 1000 steps), we deem the layers converged.

## 6.7 MIMIC-IV ENVIRONMENT

The MIMIC-IV EHR dataset was collected between 2008 and 2019 from the Beth Israel Deaconess Medical Center in Boston, Massachusetts. The latest version of this dataset, MIMIC-IV, contains de-identified data from over 524,000 distinct hospital admissions and over 257,000 patients (14). Our preprocessing steps follow the framework suggested by prior work (24; 17).

Our background dataset consists of 400 patients which healthy kidneys and our foreground dataset contains 123 patients with renal disease. Intuitively, electrolyte repletion policies must account for kidney function. For example, patients who are admitted for a renal condition have a baseline

imbalance of electrolytes and receive electrolyte repletion much differently than patients who have functioning kidneys.

Below is a list of all patient characteristics used in our experiments.

Table 2: Features used to predict electrolyte repletion.

| Feature | Description |
| --- | --- |
| Age | Age of patient at admission |
| Gender | Gender discretized to male/female |
| Expired | Binary flag indicating whether patient is dead/alive |
| Patient Weight | Patient Weight (kg) |
| Length of Stay | Length of patient's stay (days) |
| Heart Rate | Heart Rate (bpm) |
| Respiratory Rate | Rate of breathing (breaths per minute) |
| Oxygen Saturation | Oxygen saturation in blood (%) |
| Temperature | Body temperature (°F) |
| Systolic BP | Systolic Blood Pressure (mmHg) |
| Diastolic BP | Diastolic Blood Pressure (mmHg) |
| Potassium (IV) | Potassium administered through IV (mL) |
| # Hours Potassium (IV) | # hours potassium is administered through IV |
| Potassium (nonIV) | Potassium administered orally (mg) |
| # Hours Potassium (nonIV) | # hours potassium is administered orally |
| Potassium | Potassium measured in blood |
| Calcium (IV) | Calcium administered through IV (mL) |
| # Hours Calcium (IV) | # hours calcium is administered through IV |
| Calcium (non-IV) | Calcium administered orally (mg) |
| # Hours Calcium (nonIV) | # hours calcium is administered orally |
| Calcium | Calcium measured in blood |
| Phosphate (IV) | Phosphate administered through IV (mL) |
| # Hours Phosphate (IV) | # hours phosphate is administered through IV |
| Phosphate (non-IV) | Phosphate administered orally (mg) |
| # Hours Phosphate (nonIV) | # hours phosphate is administered orally |
| Phosphate | Phosphate measured in blood |
| Magnesium (IV) | Magnesium administered through IV (mL) |
| # Hours Magnesium (IV) | # hours magnesium is administered through IV |
| Magnesium (non-IV) | Magnesium administered orally (mg) |
| # Hours Magnesium (nonIV) | # hours magnesium is administered orally |
| Magnesium | Magnesium measured in blood |
| Sodium | Sodium measured in blood |
| Vasopressors | Administered to constrict blood vessels |
| Beta Blockers | Administered to reduce blood pressure |
| Calcium Blockers | Administered to reduce blood pressure |
| Loop Diuretics | Administered to treat hypertension, edema |
| Insulin | Administered to promote absorption of glucose from blood |
| Dextrose | Administered to increase blood sugar |
| Oral Nutrition | Orally administered nutrition supplements |
| Parenteral Nutrition | Non-orally administered nutrition supplements |
| Total Parenteral Nutrition | IV-based fluids used for nutrition |
| Dialysis | Binary indicator of dialysis procedure |
| Blood Transfusion | Binary indication of red blood cell transfusion |
| Coronary Artery Disease | Binary indication of disease |
| Atrial Fibrillation | Binary indication of disease |
| Congestive Heart Failure | Binary indication of disease |
| Chronic Kidney Disease | Binary indication of disease |
| Renal Disease | Binary indication of a non-chronic kidney disease |
| Paralysis | Binary indication of loss of ability to move |
| Parathyroid | Binary indication of disease |
| Rhabdomyolysis | Binary indication of disease |

Table 2: Features used to predict electrolyte repletion.

| Feature | Description |
|---|---|
| Sarcoidosis | Binary indication of disease |
| Alanine Aminotransferase | Measure in blood |
| Anion Gap | Measure in blood |
| Blood Urea Nitrogen | Measured in blood |
| Creatine Phosphokinase | Measured in blood |
| Hemoglobin | Measured in blood |
| Glucose | Measured in blood |
| Creatinine | Measured in blood |
| Lactic Acid Dehydrogenase | Measured in blood |
| White Blood Cell | Count measured in blood |

The Markov decision process (MDP) for this MIMIC-IV environment is a tuple $(\mathcal{S}, \mathcal{A}, P, R)$. Here, $\mathcal{S}$ is a 61-dimensional vector containing information corresponding to each feature in Tab. 2; $\mathcal{A} = \{[0,0], [0,10], [10,0]\}$ is the action space, each action corresponding to no repletion, low repletion and high repletion respectively; $P$ contains the (unknown) state transition probabilities guiding a patient's state dynamics during their hospital visit; and $R$ is a reward function, which returns a value based on whether the patient's observed potassium level is outside the normal potassium range and the cost of repletion. $R$ is the sum of a four dimensional vector; each element in this vector corresponds to the cost of oral repletion, the cost of intravenous repletion, a penalty for the patient's potassium level being too high, and a penalty for the patient's potassium level being too low, respectively. The reward function can be written as $r_t = w \cdot \phi_t(s_t, a_t, s_{t+1})$ where $\phi$ is a four-dimensional vector function such that:

$$\phi_t(\cdot) = \begin{bmatrix} -\mathbb{1}_{a_t{}^{route}[oral]} \\ -\mathbb{1}_{a_t{}^{route}[intravenous]} \\ -\mathbb{1}_{s_{t+1}[K]>K_{max}} \cdot 10 \left(1 + e^{-\sigma(K-K_{max}-1)}\right)^{-1} \\ -\mathbb{1}_{s_{t+1}[K]>K_{max}} \cdot 10 \left[1 - \left(1 + e^{-\sigma(K-K_{max}-1)}\right)^{-1}\right] \end{bmatrix} \in \begin{bmatrix} \{0,-1\} \\ \{0,-1\} \\ (-10,0) \\ (-10,0) \end{bmatrix}.$$

Here, $K$ is the known measurement of potassium, and $K_{max}$ and $K_{min}$ define the upper and lower bounds of the target potassium range, respectively. The first two elements of the reward vector correspond to whether the repletion was administered orally or through an intravenous line. For our experiments, we use $w = [1, 1, 1, 1]$.

