# OpenReview forum: "Nested Policy Reinforcement Learning for Clinical Decision Support"
_ICLR.cc/2022/Conference — ICLR 2022 Submitted_

### Official Review · Reviewer_rPvd · 2021-10-24

**Correctness:** 2
**Technical Novelty And Significance:** 2
**Empirical Novelty And Significance:** 1
**Recommendation:** 3
**Confidence:** 5

**Main Review:**

### **Strengths**
- Exposition is clear and easy to follow.
- Simulated experiments on CarPole are extensive and seek to answer various interesting research questions that justify the usefulness of the proposed approach.
- Good use of SHAP for understanding which features were used by the policy’s recommendation and showing the difference of foreground/background.

### **Weaknesses**
- This paper only considers two groups (background / foreground) that share state dynamics to some extent. Extensions to more than two groups could be more generally useful, but are only mentioned in future work and not discussed concretely.
- There doesn’t seem to be anything special about the proposed approach that’s specific to the offline setting. One could parameterize the Q-function as $f(s,a) = g_s(s,a) + 1_{z} g_f(s,a)$ in an online setting as well. Could you elaborate whether this approach will also provide benefit in the online setting, and if not, what makes it special for offline RL?
- The baselines were not clearly described in the main text: what is the “transfer learning” baseline (first appeared on page 4)?
- Re Real-world EHR data experiments: I don’t believe “agreement with doctor’s actions” is the right quantitative metric. Consider using an OPE (offline policy evaluation) method [1][2].
- It wasn’t clear to me whether there was a train/val/test split because Sec 4.2 on page 8 describes the total number of patients yet Table 1 says “predict … on test samples”

### **Detailed Comments**
- Re a possibly inaccurate claim on page 11 Appx 6.1: “In its original formulation, FQI was shown to converge to a Bellman-optimal Q-value function using concepts from dynamic programming theory.” The classical results only guarantees that iterative application of Bellman optimality operator (without function approximation, or equivalently, with perfect function approximation) leads to Q*. But in practice the convergence of FQI [3] depends heavily on (i) the completeness/realizability of the function class, and (ii) distribution shift in limited data. I also hope to see some theoretical support for why $f_{NFQI}(\\tilde s, a)$ is a “good” choice of function class.
- The proposed setting seems to be a variation/special case of factored MDP (with factored state spaces). I would recommend reading and citing the following references [4][5] (plus possibly their followup works) and comment on how the proposed approach is related and where it differs.
- Presentation of Fig 6: Can you please clarify what are the values in the heatmap? Since there are only 3 actions (no, low, high), I was expecting only three colors in the heatmap. I would also recommend using a discretized color bar with only 3 colors. Furthermore, the y-axis label for creatinine should be flipped so that lower values are at the bottom which is what a typical Cartesian axis looks like.
- Naming: it’s true that the presented setting has “nested” dynamics, but I’m not sure if “nested” is the best way to describe the value function and policy.

### **References**
1. Taylor W. Killian, Haoran Zhang, Jayakumar Subramanian, Mehdi Fatemi, Marzyeh Ghassemi. An Empirical Study of Representation Learning for Reinforcement Learning in Healthcare. ML4H @ NeurIPS 2020. https://arxiv.org/abs/2011.11235
2. Shengpu Tang, Jenna Wiens. Model Selection for Offline Reinforcement Learning: Practical Considerations for Healthcare Settings. MLHC 2021. https://arxiv.org/abs/2107.11003
3. Jinglin Chen, Nan Jiang. Information-Theoretic Considerations in Batch Reinforcement Learning. ICML 2019. https://arxiv.org/abs/1905.00360
4. Daphne Koller, Ronald Parr. Computing factored value functions for policies in structured MDPs. IJCAI-99. https://dl.acm.org/doi/abs/10.5555/1624312.1624408
5. Carlos Guestrin, Daphne Koller, Ronald Parr, Shobha Venkataraman. Efficient solution algorithms for factored MDPs. JAIR 2003. https://doi.org/10.1613/jair.1000


**Summary Of The Paper:**

This paper considers decision making in nested MDPs where there are two distinct groups having partially shared (but unknown) state dynamics. A new algorithm called NFQI (nested fitted-Q iteration) is presented, and it’s based on a simple modification to FQI, a commonly used offline RL algorithm. Baseline comparisons and sensitivity analyses were done in a set of simulated experiments and using OpenAI gym CartPole. The proposed approach is then applied to a real-world RL task formulated from MIMIC-IV EHR data.

**Summary Of The Review:**

The presented setting is interesting to look at and potentially useful in clinical decision making, and the simulation results justified the benefit of the proposed approach to a limited extent. However, experiments on real EHR data need to be improved, and there seems to be a lack of theoretical understanding for when and why this approach works well; this will also help generalize the results to more than two groups. There needs to be better framing regarding whether the conclusions are limited to offline settings. Lastly, it's important to include a discussion that relates to past work on factored state spaces and factored value functions.

---

### Official Review · Reviewer_KSWV · 2021-10-29

**Correctness:** 3
**Technical Novelty And Significance:** 2
**Empirical Novelty And Significance:** 2
**Recommendation:** 3
**Confidence:** 4

**Main Review:**

*Strengths*
---

The paper nicely outlines the setting for NFQI. It's clear to understand how the Nested MDPs are modeled and where they may be found in the real world. Concerns about novelty and the necessity of this formulation are listed below among the weaknesses I found in the paper.

The paper is well structured and well written for the most part. At times I felt that the discussion was a little too high level, specifically in Sections 2 and 3 (Related Work and Methods; more in the 'Weaknesses' section below) but for the most part necessary explanations were sufficiently through and clearly explained. A particular strength in the writing was evident in how the core insights or experimental objectives were clearly outlined at the beginning of Section 4 (Experiments).

I was also impressed at the extent by which the learned policies were evaluated and compared to standard FQI. The interpretation of how different features contribute to policy decisions using SHAP values was nice to see (some concerns about clarity are discussed below).

*Weaknesses*
---

I have several concerns about the necessity of this approach as proposed by the authors and presented in this paper, as submitted.

First, by focusing only on two subpopulations of these Nested MDPs the authors are investigating the simplest base case of what could be termed multi-task or meta learning. There is very little discussion to justify why taking this stripped down approach is necessary. Beyond the tendency of the multi-task or meta-learning literature to focus on large overparametrized networks, there's no reason to expect why the methods and algorithms presented in these papers wouldn't be applicable and perform as well as NFQI.

Second, there is a large body of work on transfer within RL that has been overlooked and should be cited appropriately. The Nested MDP setting seems to be a special case of a Hidden-parameter MDP (HiP-MDP) or Block MDP (references below). Each of these branches of literature cover the exact setting where families of MDPs, differentiated only by changes in their dynamics, are treated with joint policies or otherwise personalized to individual settings through a latent or contextual parameter. A discussion differentiating this paper from this body of literature is warranted.

Third, it's not surprising that NFQI outperforms FQI given the extra contextual information the algorithm is provided through the contextual group variable $z$. A more fair comparison would be providing this extra feature to FQI and evaluating the policy performance in that setting. Additionally, it's severely unclear what is actually done in the Transfer Learning baseline. Are policies learned on the background dataset and they applied directly to the foreground dataset? Is there any finetuning on the foreground at all?

Beyond these points I was disappointed with how high-level and devoid of detail the discussion in Section 3 (Methods) was. There is very little concrete development of NFQI where I wasn't entirely sure what was being presented in the experimental results and how much significance to ascribe to the performance gains or differences between FQI and NFQI. Clearly, something exciting is happening by leveraging the contextual information of the nested MDP in Equation 1 within the function approximator $f$. But without a clear link or more formal presentation of where Eqt. 1 fits into FQI to produce NFQI its not easy to immediately follow the results and insights developed in Section 4 (Experiments). It's also unclear, after reading the Appendix, what the separate Losses represent. Based on Eqt. (1) and Algorithm (1) in the appendix, it seems that $g_b$ is doubly accounted for. So is $g_b \rightarrow \pi_b$ having double the gradients applied to it? If so, it's unsurprising that it's performance is so high relative to FQI on the background dataset on the cartpole task.

**Questions from Cartpole experiments**

It's not apparent why different dataset sizes were used between the various experiments for Cartpole (also, in Section 4.1.1 does "samples" mean "episodes"?). The lack of consistency and the policies achieving different levels of performance between experiments is unnecessary and only provides more opportunity for confusion. It would possibly be better if the experiment in Section 4.1.1 had the same total number of episodes as the other experiments in Section 4.1 but still could be used to demonstrate the effects of a small portion of those episodes belonging to the foreground dataset with a more thorough exploration of this effect as is done in Section 4.1.4. What sparked this question was the diminished performance in Figure 4 in comparison to Figure 1.

The use of SHAP values to differentiate policies is a really neat idea. I think that there are further insights behind Figure 3 that probably deserve to be explored. As discussed, there are qualitative differences in the importance of features between the background and foreground policies. What is shown in addition to this is that the feature values that a deemed important are inverted between the datasets. High feature values seem to be more important in the foreground dataset. This is interesting given the definition of the foreground MDP. Does this relation shift as you reduce the magnitude of the force applied in the foreground dataset?

It's not described what's varied between runs of the experiments, are these random seeds? Sampling strategies? I.e. what is driving the variance of the performance? Is it the method itself?

In Section 4.1.4 it is mentioned that the proportion of data that makes up the foreground dataset is varied to mirror proportions seen in medical datasets. This is unecessarily vague. What types of datasets? What makes up these proportions in the medical datasets? What conditions?

**Questions from the experiments using the MIMIC renal cohort**

The cohort definition was unclear beyond the number of patients. The number of features and how they're derived is relevant information that should be included in the main body of the paper. The specific action definition is also not very clearly outlined. What goes into repleting electrolytes? Is there more than one medication for this beyond prescribing potassium? After looking through Table 2 in the appendix, it appears that several of the features are associated with treatment decisions or medications administered to the patients. There is likely some overlap with comorbidities and the presence of these medications. How was this accounted for? Were treatment/action decisions incorporated into the patient's state representation?

The potassium treatment decisions are rightfully binned into a categorical vector (none, low, high) which corresponds to three discrete actions, as I understood the definition. However in Figure 6 there seems to be a continuum of treatments that are administered. Which was it? On this point, it's not clear what the numerical values in Figure 6 correspond to in the categorical sense. This was confusing because, as stated, the healthy range for potassium is defined as 3.5-4.5 mmol/L but in Figure 6d, the NFQI policy appears to recommend 'High' replenishment treatments within that range (assuming a value of 2 = 'High'). That seems counterintuitive.

Now, for Table 1. If there are only three discrete actions, then the results of predicting the clinician's actions aren't performing much better than random. It's also curious that the action prediction performance for the background dataset is so much lower than the foreground dataset. Perhaps this is due to greater heterogeneity in the background data? I would expect the performance to be better on the background dataset due to there being more data.

In Figure 7, it's interesting to see how the foreground and background policies differ from each other. I'm curious if there's some causal leakage informing these results however. For example, one of the top rated features is Potassium, which is directly correlated to the treatment decisions, right? Is this filtering through as a top rated feature in the foreground dataset because there is a higher frequency of these labs/tests being ordered? Is this the same for HGB or the amount of phosphate being administered via IV? What is the significance of adding the other features together. This imbalances the presentation of Figure 7 and my immediate impression was that the other 56 features are more important since their combined SHAP values are significantly higher than the others presented. I would perhaps recommend removing that last row and making a comment in the caption (or main body) that the top 5-10 features from NFQI are provided...


----
`References`

**HiP-MDPs**

Doshi-Velez, F., & Konidaris, G. (2016, July). Hidden parameter markov decision processes: A semiparametric regression approach for discovering latent task parametrizations. In IJCAI: proceedings of the conference (Vol. 2016, p. 1432). NIH Public Access.

Killian, T., Daulton, S., Konidaris, G., & Doshi-Velez, F. (2017, December). Robust and efficient transfer learning with hidden parameter Markov decision processes. In Proceedings of the 31st International Conference on Neural Information Processing Systems (pp. 6251-6262).

Yao, J., Killian, T., Konidaris, G., & Doshi-Velez, F. (2018). Direct policy transfer via hidden parameter markov decision processes. In LLARLA Workshop, FAIM (Vol. 2018).

Perez, C., Such, F. P., & Karaletsos, T. (2020, April). Generalized hidden parameter mdps: Transferable model-based rl in a handful of trials. In Proceedings of the AAAI Conference on Artificial Intelligence (Vol. 34, No. 04, pp. 5403-5411).

**Block MDPs**

Misra, D., Henaff, M., Krishnamurthy, A., & Langford, J. (2020, November). Kinematic state abstraction and provably efficient rich-observation reinforcement learning. In International conference on machine learning (pp. 6961-6971). PMLR.

Zhang, A., Lyle, C., Sodhani, S., Filos, A., Kwiatkowska, M., Pineau, J., ... & Precup, D. (2020, November). Invariant causal prediction for block mdps. In International Conference on Machine Learning (pp. 11214-11224). PMLR.

Zhang, A., Sodhani, S., Khetarpal, K., & Pineau, J. (2020). Learning Robust State Abstractions for Hidden-Parameter Block MDPs. arXiv preprint arXiv:2007.07206.

Zhang, A., Sodhani, S., Khetarpal, K., & Pineau, J. (2020). Multi-Task Reinforcement Learning as a Hidden-Parameter Block MDP. arXiv e-prints, arXiv-2007.

**Summary Of The Paper:**

This paper introduces a learning paradigm to handle related MDPs (called Nested MDPs) that share the same structure and definition, varying only in their dynamics. Based on Fitted Q-Iteration (FQI), the authors propose an algorithm known as Nested FQI (NFQI) to learn from the shared structure of the Nested MDPs while also being able to adapt to the specific dynamics of the separate MDPs. NFQI is developed and analyzed empirically in the simplest setting where there are only two variants of dynamics are present, where there is a nominal imbalance between expected and out of distribution observations. NFQI is compared to standard FQI and standard transfer learning on an augmented Cartpole task as well as medical treatment task derived from retrospective EHR data. Extensive analysis on the learned policies is performed to establish the anticipated benefits of using NFQI in these settings.

**Summary Of The Review:**

There are several concerns about significance of the problem setting and proposed algorithmic approach. Additionally, there are severe gaps in clear exposition outlining how the proposed algorithm is set-up and run.

---

### Official Review · Reviewer_gMUj · 2021-10-30

**Correctness:** 2
**Technical Novelty And Significance:** 2
**Empirical Novelty And Significance:** 1
**Recommendation:** 3
**Confidence:** 4

**Main Review:**

Strengths:
- The challenge of being able to identify shared problem structure between subgroups and leverage that information to improve sample efficiency in reinforcement learning is an important and difficult task. This work provides an interesting perspective and intuitive approach for solving it.
- I thought the questions and experiments laid out in Section 4 (top of page 4) were interesting and well-defined. Overall, their experimental design(s) in the CartPole setting were compelling.

Weaknesses:
- My main critique of the paper is that it takes an unnecessarily complicated approach to an otherwise easy problem and does not compare performance of the proposed algorithm to one of the more obvious/intuitive benchmark approaches. First, I'll address the "unnecessarily complicated approach" comment. The proposed approach is to essentially create two separate sets of weights for modeling a group-specific $Q$-value estimator: one shared set of weights for all groups, and one set of weights unique to the "foreground" group. But the authors themselves state that their approach is equivalent to just augmenting the state space with an indicator of one-hot encoded group assignment. In this light, an alternative summary of the paper's contributions could be written as, "We find that FQI performs better when it has access to all of the variables that actually specify the MDP". Fundamentally, when group assignment modifies the MDP but is not included in the state space, the resulting stochastic decision process is no longer Markovian. I suppose you could think of the resulting problem setting as a POMDP. The resulting problem is interesting when group assignment is not observable (see e.g., Killian and Daulton [1]), but trivial when group assignment is available: just expand the state space to include the state space and the problem is solved. In my mind, then, to the extent that NFQI is not always just a simple state space augmentation (see bullet below), the best comparison here would just be FQI with an augmented state space. I would bet, though, that any differences in performance between NFQI and state-augmented FQI (when there is actually a difference between them) are negligible at best. The other baseline to compare against in class-imbalanced settings would be to use FQI + augmented state space + sample weighting to address the class imbalance (so upweighting the loss of the smaller class by ~1/P(a random sample is in the smaller class)).
- Related to the above, I'll note that the authors' comment in Appendix 6.1 about equivalence of their proposed $f_{\text{NFQI}}$ with state-augmented FQI is _not_ true for the neural network approach that they describe in Appendix 6.2. This discrepancy is not mentioned, but should be. More importantly, however, the authors gloss over the fact that convergence of FQI (see Section 3.5 of [2]) is heavily dependent on the use of an induced kernel that does not depend on the output values of the training sample. This is certainly not the case when using neural networks as function approximators, as the authors have proposed. Indeed, _none_ of the instantiations of NFQI proposed by the authors carry such a convergence guarantee.
- In the discussion of the related work, the authors claim that their approach differs from Meta-RL and Multi-task RL by arguing that these approaches deal with different tasks _in addition to_ different MDPs, but the NFQI approach only handles modeling the _same_ task with different MDPs. I would argue that Multi-task RL and Meta RL are thus more general than the proposed NFQI approach and in fact subsume it. To that extent, one relevant comparison here would be to compare Multi-task RL and Meta-RL approaches to the proposed NFQI approach. But honestly, this is probably overkill because with known group identity you can likely get most of the potential gains just by augmenting the state space and running FQI.
- I have significant concerns about the quantitative evaluation approach for the MIMIC dataset/task. Using alignment between clinician behavior and the learned RL policy as a proxy for policy quality is fraught in many ways (What if clinicians behave suboptimally? What if the differences are just due to chance?). See Gottesman et al [3] for some pointers to best practices in off-policy policy evaluation.
- Two central premises of the proposed approach is that there are many important real-world problem settings in which (assuming some shared RL task/reward function), (1) there are precisely two known/observable conditions for which the underlying transition dynamics of the associated MDP are similar but slightly different; and (2) there are significantly more data available for one condition relative to the other.
- While premise (1) is plausible, it is poorly (if at all) justified. The much more likely scenario in my mind is one in which there are _many_ conditions/groups, all of which are _unobservable_, each of which is associated with a slightly altered MDP. Consider the experiment the authors ran on MIMIC, where they split the cohort into individuals with kidney disease and individuals without kidney disease. Who's to say that this is the right way to split that cohort into two groups? Would it be more appropriate to split patients on age? Diabetes status? Prior history of an Acute Kidney Injury? Also, does the specific kind of kidney disease matter? What about the severity? Is there some grouping that is unobserved/latent but perhaps correlated with age, diabetes status, and kidney disease that would be a more accurate description of how patient-specific MDPs vary? The limitation of the presently proposed framework to exactly two observed groups is quite constraining and limits the significance of the work. That being said, in the spirit of "All models are wrong, but some are useful", I'll put that aside for now and focus on the merits of the approach for improving performance in the proposed problem setting.
- Premise (2) is also plausible, but also poorly justified. The justification for this claim is essentially an intuitive argument in the second sentence of Section 4.1.4 "NFQI is robust to group size imbalance": "For example, it is much easier to collect medical records from healthy (background) patients than from patients with a specific, chronic disease". I would actually argue that the opposite is true in the medical setting. Medical record data for healthy individuals are relatively sparse because healthy patients don't go to the hospital. On the other hand, individuals with chronic conditions will tend to visit their primary care provider (PCP) or specialist with much more regularity. I'm not arguing with the premise itself, but rather saying it needs more justification and the one given is inadequate (and, in my view, just wrong).

I'm not going to get into really minor points here, because I think the paper will require a major overhaul before it can be accepted at a conference like ICLR. That being said, I do think there are some interesting directions here for future work, specifically in settings where there is more than one group and group identity is unknown. There's already a fair bit of existing work on Meta RL and Multi-task RL that is quite relevant. Another interesting direction on the theory side would be to better understand how "close" MDPs from two or more groups have to be in order for weight-sharing approaches like the one proposed provide sample efficiency improvements.

[1] Killian, Taylor, et al. "Robust and efficient transfer learning with hidden parameter Markov decision processes." Proceedings of the 31st International Conference on Neural Information Processing Systems. 2017.

[2] Ernst, Damien, Pierre Geurts, and Louis Wehenkel. "Tree-based batch mode reinforcement learning." Journal of Machine Learning Research 6 (2005): 503-556.

[3] Gottesman, Omer, et al. "Evaluating reinforcement learning algorithms in observational health settings." arXiv preprint arXiv:1805.12298 (2018).

**Summary Of The Paper:**

This paper proposes nested policy fitted Q-iteration (NFQI), an adaptation of the well-established Fitted Q-Iteration (FQI) algorithm originally proposed by Ernst, Geurts, and Wehenkel in 2005 [1] to the setting where the agent is required to learn policies for two different MDPs while sharing information between them. At a high level, the authors' proposed modification involves learning a function $f$ which approximates the Q-function, $Q^{\pi}(s_t, a_t) = r_t + \gamma \mathbb{E}_{s'}[V^{\pi}(s')]$ by decomposing the $Q$-estimate into a shared component and a subgroup-specific component.

Concretely, they represent $f$ as $f(\mathbf{s}, \mathbf{a}, z) = g_s(\mathbf{s}, \mathbf{a}) + \mathbb{1}[z=1]g_f(\mathbf{s}, \mathbf{a})$ where $g_s$ models the $Q$-value using parameters shared across groups and $g_f$ models a group-specific modification to the estimated $Q$-value.

Through both simulated experiments (CartPole) and an observational dataset (electrolyte replenishment optimization on MIMIC) the authors demonstrate that their NFQI approach performs better than do alternative approaches which do not take known subject-specific group information into account.

[1] Ernst, Damien, Pierre Geurts, and Louis Wehenkel. "Tree-based batch mode reinforcement learning." Journal of Machine Learning Research 6 (2005): 503-556.

**Summary Of The Review:**

This paper proposes an approach for intelligent weight sharing in offline RL when the dataset available consists of two groups with slightly different MDPs, and where group identity is known/observable. This problem can be trivially solved by simply including group identity in the state representation. The experimental design is good and the research questions are interesting. However, the problem is simple enough and the solution trivial enough that there's not a whole lot of novelty or significance in this paper. I recommend rejection.

---

### Official Review · Reviewer_EMFG · 2021-11-07

**Correctness:** 3
**Technical Novelty And Significance:** 2
**Empirical Novelty And Significance:** 2
**Recommendation:** 5
**Confidence:** 2

**Main Review:**

Pros:
- A simple approach to account for a structured and imbalanced data when learning off-policy RL
- NFQI works well in the case of extreme imbalance between the foreground and background environments
- The interpretability experiments using SHAP values validate that the NFQI finds policies that identify the foreground environment as an important component
- The model is applied to real-world clinical data (MIMIC-IV)


Cons:
- The novelty of NFQI is somewhat limited. The proposed training approach is very similar to transfer learning, where  shared and specific components are trained.
- Limited baselines in the experimental analysis



**Summary Of The Paper:**

The paper proposes an off-policy RL called NFQI (Nested Fitted Q-Iteration) as an extension of Fitted Q-iteration to estimate group-specific policies and account for group structure in the data having two pre-defined groups of observations (background and foreground). NFQI imposes a structure on the family of function to approximate the Q function with guarantees to converge to Bellman optimal Q-value function. NFQI is trained in two stages: The first stage the shared component is trained with all samples. In the second stage, the foreground component is trained using only the foreground samples. The NFQI method is validated on a nested cart-pole environment where the background is the original environment and foreground includes a constant force that pushes the cart to the left. NFQI is also validated using a real-world clinical data from MIMIC dataset.

**Summary Of The Review:**

NFQI has many real-world applications, including the use in healthcare. However, the novelty of the method is rather limited. Also, comparisons with additional baselines would be required to further validate the approachg.

Minor comments:
- background loss L_b is not defined in Algorithm 1
- Table 2 is duplicated, so which list of features has been used in the experiments shown in the paper ?
- In figure 5-a superimposed bars are not clear as mixed color is showing up. Separate bars would be better for clarity.

---

### Decision · Program_Chairs · 2022-01-20

**Decision:**

Reject

**Comment:**

This paper provides a method for offline RL in settings where the environment may exhibit significant similar structure, such as one part having nearly the same dynamics as other parts. The work is motivated in part by healthcare settings. The reviewers appreciated the potential applications to areas like healthcare but also thought there is a strong body of related work (e.g. transfer learning, meta-RL and other related papers) and it was unclear how novel the approach was within that related work, or how it would compare. The authors did not respond to the reviewers’ reviews. We hope their input is useful to the authors’ in revising their work for the future.